# Explicit Characterization of Spatial Heterogeneity Based on Water Quality, Sediment Contamination, and Ichthyofauna in a Riverine-to-Coastal Zone

**DOI:** 10.3390/ijerph16030409

**Published:** 2019-01-31

**Authors:** Dong-Kyun Kim, Hyunbin Jo, Inwoo Han, Ihn-Sil Kwak

**Affiliations:** 1Fisheries Science Institute, Chonnam National University, Yeosu 59626, Korea; dkkim1004@gmail.com (D.-K.K.); prozeva@chonnam.ac.kr (H.J.); 2Faculty of Marine Technology, Chonnam National University, Yeosu 59626, Korea; haninwoo89@gmail.com

**Keywords:** coastal bay, environmental assessment, sediment contaminant, Self-Organizing Map, water quality

## Abstract

Our study aims to identify the spatial characteristics of water quality and sediment conditions in relation to fisheries resources, since the productivity of fisheries resources is closely related to the ambient conditions of the resource areas. We collected water quality samples and sediment contaminants from twenty-one sites at Gwangyang Bay, Korea, in the summer of 2018. Our study sites covered the area from the Seomjin River estuary to the inner and outer bays. To spatially characterize physicochemical features of Gwangyang Bay, we used Self-Organizing Map (SOM), which is known as a robust and powerful tool of unsupervised neural networks for pattern recognition. The present environmental conditions of Gwangyang Bay were spatially characterized according to four different attributes of water quality and sediment contamination. From the results, we put emphasis on several interesting points: (i) the SOM manifests the dominant physicochemical attributes of each geographical zone associated with the patterns of water quality and sediment contamination; (ii) fish populations appear to be closely associated with their food sources (e.g., shrimps and crabs) as well as the ambient physicochemical conditions; and (iii) in the context of public health and ecosystem services, the SOM result can potentially offer guidance for fish consumption associated with sediment heavy metal contamination. The present study may have limitations in representing general features of Gwangyang Bay, given the inability of snapshot data to characterize a complex ecosystem. In this regard, consistent sampling and investigation are needed to capture spatial variation and to delineate the temporal dynamics of water quality, sediment contamination, and fish populations. However, the SOM application is helpful and useful as a first approximation of an environmental assessment for the effective management of fisheries resources.

## 1. Introduction

Coastal ecosystems typically retain high economic and environmental values in light of aquatic resources and biodiversity [1]. Many coastal areas play an important role in operating fisheries and related industries around the world [2]. In East Asia, fisheries resources are at the core of the marine industry in several countries including South Korea, East China, and Japan; this is due to the high biodiversity and high productivity in the coastal regions [3,4]. Biodiversity and productivity are closely linked to biogeochemical processes among water column, sediment, and biological entities. This complicated ecological interplay of the coastal ecosystem can respond differently to the surrounding environmental conditions, which are driven by anthropogenic disturbances as well as climatic changes [5]. In this regard, ambient aquatic pollution may be a critical factor determining biodiversity and productivity.

The productivity of coastal ecosystems, such as primary production and fish biomass, intertwines with the multiple factors/functions containing nutrient concentrations (carbon, nitrogen, and phosphorus), phytoplankton growth, zooplankton grazing effects, and benthic biota (e.g., crustaceans and shellfish). In particular, the ambient water quality can have a significant effect on changes in the microbial food web structure and functions, which involve the biological community composition, hypolimnetic hypoxia/anoxia, and toxic algal blooms in coastal areas [6,7,8,9,10]. On top of the microbial food web changes, sediment contamination is another causal factor affecting the benthic biota, a significant food source for fish [11,12,13,14]. Additionally, resuspension of sediment micro-organisms into the water column can increase the health risk for those who use water [15]. It is often reported that high concentrations of heavy metals in sediment can lead to their accumulation in the internal organs of crustaceans and shellfish [16]. Moreover, heavy metals are more stable and persistent than other contaminants of aquatic ecosystems [17]. Therefore, identifying the pattern of sediment contamination in space and/or time is essential in order to assess fish population dynamics in coastal areas.

However, the nonlinearity and complexity of environmental monitoring data often causes difficulties when analyzing spatial and temporal patterns of ecological phenomena. Since conventional statistical methods infer cause and effect under the assumption of a linear relationship between the two, they are limited when it comes to capturing the nonlinear patterns of ecological phenomena. Conventional data-ordination methods such as principal components analysis (PCA) and correspondence analysis encompass the main drawback of distortion and artifact effects known as ‘*horseshoe effect*’ or ‘*arch effect*’, although they have been widely used for characterizing complex features of interest [18]. It has recently been reported that a linear multivariate analysis could not explicitly account for the complex ecological interplay between phytoplankton, macrophytes, and sediment nutrient release [19]. In contrast, several studies have shown that nonlinear methods, such as artificial neural networks, are more suitable for global sensitivity analysis than multiple-linear regression [19,20]. Specifically, Kim et al. [19] identified an abrupt regime shift of an ecosystem, from a turbid phytoplankton-dominant state to a clear macrophyte-dominant state, using Self-Organizing Map (SOM), which is a type of unsupervised artificial neural network. Astel et al. [21] emphasized the power of the classification by SOM over traditional methods such as PCA. Recently, SOM applications have been expanded to water quality patterns associated with land use and socio-environmental management [22,23,24]. Therefore, given the complex dynamics of ecosystems, nonlinear analytics could be highly successful in characterizing the biogeochemical features of coastal bays.

The main objective of our study is to identify the spatial pattern of water quality and sediment conditions at Gwangyang Bay in South Korea. We apply a nonlinear modeling method, Self-Organizing Map, for the rigorous assessment of spatial patterns in water quality and sediment contamination. According to the spatial characterization, we then discuss the fish communities in the bay in terms of fisheries resources based on the results. The present study could be novel in view of the expansion of SOM applications for understanding biological communities associated with ambient physicochemical conditions. Furthermore, our assessment will be valuable as a first approximation of spatial heterogeneity in limno-oceanographic patterns for effective management of fisheries resources.

## 2. Materials and Methods 

### 2.1. Description of the Study Area

Gwangyang Bay is part of the Korean National Archipelagos that are located off the south coast of the Korean peninsula (Figure 1). The bay receives an annual mean discharge of 2298 × 10^6^ m^3^ yr^−1^ from the Seomjin River [25]. A significant amount of nutrients drains into the system from the watershed (ca. 5000 km^2^). The water depth varies from 10 m at the Seomjin River estuary to 50 m at the outer Gwangyang Bay. The tidal cycle appears to be semi-diurnal. Compared to other Korean river estuaries that have barrages, the Seomjin River estuary remains open, and thus the water mass is exchanged between the river and ocean more actively. The natural condition of Gwangyang Bay remains highly productive as well as biologically diverse. In this respect, Gwangyang Bay (ca. 450 km^2^ from the estuary to the outer bay) is the most economically productive in Korea. Specifically, amongst three metropolitan cities and eight provinces, Jeonnam Province with Gwangyang Bay provided 71% (1,297,815 tons per year) of the aquacultural resources, as of 2016 [26]. On the other hand, there is a large industrial area near the bay, and the area is primarily involved with oil refineries and steel plants. This industrial area is regarded as a significant point source of chemical contaminants to the bay. In addition, intermittent spills from large oil tankers are another factor disturbing the water quality and benthic sediment [27].

### 2.2. Sampling and Data Collection

The survey was conducted for three days in June 20–22, 2018 when a neap tide was formed. We sampled water at a depth between 0 m and 5 m using a Van Dorn sampler (horizontal PVC Alpha water sampler, size: 8.2 L). Particularly for sampling water adjacent to the river mouth, we sampled water at a depth between 0 m and 1 m due to the shallow depth relative to the surrounding ocean. The benthic sediments were collected using an Ekman sampler (M197-C12 manufactured by Wildlife Supply, Yulee, FL 32034, USA, size: 23 × 23 × 23cm), and then were stored in a 200-ml polyethylene container. A total of twenty-one sampling sites were selected, which covered an extensive area from the Seomjin River estuary to the outer Gwangyang Bay (Figure 1). The first day’s survey was based on six sites (1–6 in Figure 1), the second day was based on eight sites (7–14 in Figure 1), and the last day was based on seven sites (15–21 in Figure 1). The water temperature and salinity were measured on site using portable equipment (YSI Professional Plus, YSI Inc., OH 45387, USA). Since the samples were based on surface water sampling, their vertical gradient was not investigated. The nutrient and chlorophyll-*a* concentrations were analyzed from the water samples in the laboratory. Those concentrations were based on spectrophotometry (OPTIZEN POP Series UV-Vis, KLAB Daejeon, Korea). Particularly for chlorophyll-*a* measurement, the water samples were filtered through a 0.45 μm pore-size membrane (Advantec MFS membrane filter, Dublin, CA 94568, USA); the membrane was then homogenized prior to the spectrophotometry (OPTIZEN POP Series UV-Vis). Their concentrations were measured according to standard analytical methods proposed by the Korean Ministry of Oceans and Fisheries [28] and the Korean Ministry of Environment [29]. Organic and inorganic carbon concentrations were measured using a carbon analyzer (vario TOC cube, Elementar Americas Inc., NY 11779, USA) according to 850 °C combustion catalytic oxidation methods. A total of twelve sediment contaminants (Ni, Zn, Co, Se, Fe, Al, As, Cd, Pb, Cr, Cu, and Mn) were measured. For preprocessing, the sediment samples (0.5 g from the 200-ml sample; the aliquot was weighed with precision to a hundredth of a milligram) were digested by adding a mixture of 65% nitric acid (HNO_3_), 65% perchloric acid (HClO_4_), and 30% hydrogen peroxide (H_2_O_2_). The acidic digestion of the samples was then processed using a microwave decomposition system. After cooling, the samples were added to 0.1% nitric acid. We measured the sediment contaminants using an inductively-coupled plasma mass spectrophotometer (NexION/300X, PerkinElmer Inc., MA 02451, USA). Regarding the survey for ichthyofauna, beam-trawling was implemented to collect fish samples with respect to four sites (8, 11, 12, and 13) on July 1–2, 2018. The catching/towing time was 30 minutes at each site (net width: 8 m, 1.9-cm mesh-sized wing and body, and trawling distance: 2 km). The survey was limited to the four aforementioned sites due to several issues of safety related to the route of industrial ships, as well as the protection of fisheries resources, at the 17 other sites. The fish samples were immediately weighed and identified on site. The identification of the fish species was conducted according to taxonomy by Aizawa et al. [30] and Yoon [31].

### 2.3. Application of Machine Learning to the Data Analysis

Machine learning is an area of artificial intelligence that enables an algorithm to learn from data, thereby extracting information by incorporating adaptive and self-organizing properties [32]. Amongst the current machine learning algorithms, artificial neural networks (ANNs) have been widely used to search for optimal solutions using such data-learning processes. Specifically, SOM is a type of unsupervised ANN, and it is a powerful tool for the recognition of meaningful patterns and features in complex data (Figure 2). SOM is capable of reducing multi-dimensional scales and mapping into two-dimensional planes [33]. In ecological research, SOM is now considered to be more appropriate for multivariate analysis than other conventional statistical techniques [34]. Several studies have shown that the use of SOM is a robust way to capture the nonlinear pattern of ecosystems [19,20]. For these reasons, SOM has been extensively applied to pattern recognition in various ecological domains, including benthic macroinvertebrates [35,36], plankton communities [37,38], and biomanipulation assessments [39].

The SOM was trained using nineteen input variables including seven water quality parameters (water temperature (WT), salinity, total phosphorus (TP), total, nitrogen (TN), total organic carbon (TOC), total inorganic carbon (TIC), and chlorophyll *a* (Chl-*a*)) and twelve sediment contaminants (Ni, Zn, Co, Se, Fe, Al, As, Cd, Pb, Cr, Cu, and Mn). The SOM size (i.e., the number of hexagons in the map) was determined based on the rule of $5\sqrt{\mathrm{sample}\;\mathrm{size}}$ [40]. To cluster the SOM map, we applied a hierarchical cluster analysis using Ward’s linkage method [41]. The SOM model was developed using MATLAB 6.1 (The MathWorks Inc., Natick, MA, USA) and the SOM Toolbox (Helsinki University of Technology, Helsinki, Finland).

## 3. Results

### 3.1. Features of Surface Water Quality at Gwangyang Bay

Water temperature was in the range between 22 °C and 26 °C at Gwangyang Bay in June (Figure 3). Overall, lower water temperatures were recorded in the outer bay than in the estuary and inner bay. However, we also found a decrease in water temperature at a few sites around the inner bay, including the sites 4, 5, and 6. In addition, it is notable that there was a subtle difference (~1 °C) in water temperature between site 11 and site 12, which are geographically close to each other. 

The salinity pattern presents the extent to which the freshwater of rivers affects the coastal bay. Given minimum salinity across the bay, sea water basically affects all of the study sites (Figure 3). The relatively low salinity values, observed from the sites 1, 2, and 6, were still as high as sea or brackish water (>15 psu). In the outer bay (the sites 13–21), the salinity values were high and stable at approximately 29 psu. 

From the distributional pattern of the phosphorus concentration, we found a clear decline in the concentration away from the estuary (sites 1, 2, and 3) towards the outer bay (sites 13–21) (Figure 3). The TP concentrations were high near the estuary (ca. 0.1 mg TP L^−1^). In the outer bay, TP was relatively low, but exceeded 0.05 mg L^−1^ (average of 0.075 mg L^−1^), which is typically considered to be eutrophic conditions [42,43]. In the inner bay, the TP concentrations were intermediate between those in the estuary and the outer bay; however, there was some spatial variation within the inner bay (sites 6–10). Similar to the TP concentrations, the TN concentration also decreased away from the estuary towards the outer bay (Figure 3). Compared to TP, the pattern of decline is relatively clear in TN. It is notable that the shape of the TN concentrations was mesotrophic (average of 0.365 mg L^−1^) in the range of 0.2 to 0.8 mg L^−1^ [43].

In contrast to phosphorus and nitrogen, the concentration of carbon increased away from the estuary towards the outer bay (Figure 3). Inorganic carbon comprised approximately 90% of the total carbon (TC), and thus the TIC pattern was analogous to that of the TC. It was difficult to identify the spatial and longitudinal gradient of the TOC concentrations. It appears that the TOC concentrations were relatively low in the inner bay. We also compared these results with the chlorophyll-*a* concentrations, assuming that a large portion of TOC in the water column was arising from phytoplankton biomass. However, there was no relationship between the TOC and the Chl-*a*. The Chl-*a* concentrations were mostly low (<5–7 μg L^−1^, average of 4.69 μg L^−1^), but at site 7 the Chl-*a* concentration was significantly higher (13.9 μg L^−1^). It was remarkable that the highest value of carbon concentrations was observed in inorganic form (TIC), not organic (TOC) which directly replects Chl-*a* concentration at site 7 (Figure 3).

### 3.2. Distribution of Sediment Contaminants in the Bay 

Overall, the sediment contaminants showed spatial variation in their concentrations. Compared to the water quality parameters (e.g., water temperature, salinity, phosphorus, nitrogen, and carbon), the concentration of several metal contaminants, including nickel (Ni), cobalt (Co), iron (Fe), and lead (Pb), showed a similar spatial distribution (Figure 4). At sites 2, 10, and 11 in particular, the concentrations of these contaminants were relatively low. Zinc (Zn), selenium (Se), aluminum (Al), arsenic (As), chrome (Cr), copper (Cu), and manganese (Mn) also shared spatial commonality to some extent (Figure 4). Compared to Ni, Co, Fe, and Pb, these heavy metal contaminants exhibited higher concentrations to some extent in the inner bay (sites 4–7). It is notable that cadmium (Cd) presented the highest concentration in the estuary (at site 1); excluding this site, the spatial concentrations of cadmium were slightly higher in the inner bay than in the outer bay. In summary, there is spatial variation in the heavy metal contamination of sediment; however, it was quite difficult to quantitatively distinguish the characterization of sediment contamination in space based merely on the visual comparison of bar graphs.

### 3.3. Spatial Characterization of Physicochemical Attributes Using Self-Organizing Map

The SOM analysis provided a different visualization for characterizing the physicochemical attributes of Gwangyang Bay (Figure 5). Unlike the previous comparison based on bar graphs (Figure 3 and Figure 4), the SOM simultaneously extracted a comparative pattern of both water quality and sediment contaminants across the bay. We could explicitly delineate the concentrations of physicochemical components, comparing the contrast of the gradient on a black and white scale (Figure 5). At Gwangyang Bay, in common with typical coastal ecosystems, higher water temperatures corresponded to lower salinity, and vice versa. The phosphorus and nitrogen concentrations presented nearly identical SOM patterns; this means that these nutrient patterns were spatially similar across the bay. Interestingly, we observed the inverse relationship between carbon and other nutrients (i.e., phosphorus and nitrogen), as found from the earlier graphs (Figure 3). This inverse relationship was clearer for TIC versus either TP or TN. It appears that higher Chl-*a* concentrations corresponded to lower levels of TOC. Regarding the sediment contamination, the SOM also provided heavy metal patterns in comparison to the ambient water quality. Most of the sediment contaminants, except cadmium, exhibited high concentrations in the bottom-right of the SOMs, which implies that there are a group of sites at Gwangyang Bay where heavy metals concentrations are high (Figure 5). On the contrary, the top-left of the SOMs represented a group of sites showing low concentrations of the sediment contaminants. Consequently, it was found that the spatial distribution of water quality and sediment contamination were clearly separated at Gwangyang Bay. 

Following the previous separation, the SOM finally classified four groups of physicochemical attributes at the bay, producing a pattern of the spatial characteristics of water quality and sediment contamination (see the bottom-right corner of Figure 5). Group 1 represented the main channel of the bay (sites 8, 10, and 11); Group 2 mostly belonged to the outer bay (sites 12–21); Group 3 corresponded to the estuary (sites 1 and 2); and Group 4 covered the inner bay (sites 3–7, and 9). To identify the main characteristics of water quality and sediment contamination in each group, the average values of the variables were compared (Table 1 and Figure 6). Group 1 (referred to as the main channel) was dominated by low concentrations of sediment contaminants. Group 2 (referred to as the outer bay) exhibited higher saline conditions along with high concentrations of TIC, nickel, cobalt, and iron. Group 3 (referred to as the estuary) showed high concentrations of nutrients (phosphorus, nitrogen, and organic carbon), and the salinity was distinctly lower than in the other groups. The sediment appeared to be intermediately contaminated with heavy metals, while iron and cadmium showed the highest concentrations among the groups. Group 4 (referred to as the inner bay) appeared to be the most contaminated by heavy metals, such as zinc, selenium, aluminum, arsenic, lead, chrome, copper, and manganese. Moreover, it is remarkable that the Chl-*a* concentration was also the highest in Group 4. From the SOMs, Gwangyang Bay can be characterized according to four distinct groups of physicochemical attributes in accordance with geographical zonation, such as the estuary, inner bay, main channel, and outer bay.

## 4. Discussion

In this study, we successfully characterized the dominant pattern of physicochemical attributes at Gwangyang Bay using SOM. Compared to visualization using bar graphs (Figure 3 and Figure 4), the SOM was more convenient for detecting the magnitude of the variables, as well as revealing the relationships between them. In fact, the SOM is a very efficient algorithm for comprehensively presenting complex data. For this reason, its application has gradually been increasing with respect to complex biological domains such as plankton dynamics [37,38,44], fish communities [45,46,47], and benthic macroinvertebrates [48,49,50]. Conventional multivariate methods, including principal component analysis and correspondence analysis, are limited in their ability to extract nonlinear features of a complex dynamic nature [34,51]. However, the property of SOMs that mimics the principle of competition and adaptation inherent in biological systems is better suited to nonlinear relationships in data processing [52]. Although an earlier modeling study compared SOM with other conventional statistical methods [34], recent work has also demonstrated the robustness of its application to the evaluation of abrupt/nonlinear ecological regime shifts [19]. In this respect, we highlight the fact that SOM is an effective analytical tool for characterizing ecological attributes, explicitly mapping the relationships between parameters/variables (Figure 5).

The robustness and power of SOMs pertinent to multivariate ecological analysis has enabled us to understand and elucidate the spatial pattern of water quality and sediment contamination associated with the surrounding natural and anthropogenic conditions at Gwangyang Bay. The four groups of physicochemical attributes reflect the unique characteristics of geographical zonation from river to bay (i.e., estuary, main channel of the bay, inner bay, and outer bay) (Table 1 and Figure 5). Consequently, the SOM result manifests the dominant physicochemical attributes of each geographical zone associated with the patterns of water quality and sediment contamination. 

First, one of the dominant physicochemical attributes was the nutrient gradient from Seomjin River to Gwangyang Bay. It is well known that rivers are a significant source of nutrients to coastal areas [53,54]. As such, high concentrations of phosphorus and nitrogen were found in Group 3 of the SOM (referred to as the estuary) (Table 1 and Figure 6). Additionally, the longitudinal gradient of salinity related to the nutrient pattern was evident in accordance with the nutrient gradient (Figure 3). In fact, numerous studies have produced similar results. Baek et al. [55] reported that surface nitrate concentrations were increased by Seomjin River runoff and inversely correlated with salinity at Gwangyang Bay. A similar pattern has been reported in other international studies as well [53,54]. In the Yangtze River Estuary, which is geographically close to Gwangyang Bay in climatic scale, phosphorus and nitrogen were found to occupy >60% of the total nutrient flux [56]. A large amount of nutrients coming from the river may potentially affect biomass of (generic or specific) fish communities via bottom-up food chain flow.

Second, another interesting pattern was that the main channel of the bay showed relatively low concentrations of sediment contaminants (Table 1 and Figure 6). Given that heavy metal contaminants stem from exogenous sources of chemical pollutants (i.e., the industrial complex adjacent to the bay), we conjecture that the relatively fast current and hydrodynamics in the middle of the bay could reduce the quantity of heavy metal contaminants settling into the sediments [57]. On the contrary, the inner bay featured high concentrations of heavy metals, which implies that the relatively enclosed shape of the bay could lead to the accumulation of these metals. Nevertheless, sediment contamination mostly remained within acceptable levels based on the pollution index [13]. Counter to this judgement, the inner bay could be considered contaminated based on the concentrations of Zn, Pb, Cr, Cu, and Mn, if using different criteria such as the Marine Sediment Pollution Index [58]. Several studies have assessed heavy metal concentrations of sediment, benthos, and fish in rivers and estuaries [59,60]. For example, in another part of Asia, the Tigris River’s sediments showed twice the heavy metal concentrations found at Gwangyang Bay [16]. The same study reported that certain levels of heavy metals were detected in some fish species. Specifically, *Silurus triostegus*, *Mastacembelus simacks*, and *Mystus halepensis* accumulated heavy metals in their liver, gill, and muscle tissues (Cu: 2–5%, Ni: 0.6–1.0%, Mn: 0.2–1.6%, Fe: 0.1–0.7%; % concentration = mg kg^−1^ in wet weight/mg kg^−1^ in dry weight; Tables III–VI by Karadede-Akin and Unlu, [16]). In this context, there have been public health concerns in relation to the heavy metal contamination of sediment. Specifically, Yi et al. [60] reported that human health can be potentially influenced by eating fish that were affected by their benthic food sources and sediment contamination with respect to the lower Yangtze River. Thinking along the same lines, we presume that a certain amount of heavy metal could be transported from sediment to fish via benthic food chain flow at Gwangyang Bay. The characterization of sediment contamination using SOM indicates that intense monitoring and bio-assessment are further required for the inner bay. If commercial fish were exposed to the heavy metal contamination (even though it is below acceptable ranges of contamination), the advisory criteria for fisheries resources would be needed. To this end, the SOM result can offer guidance for fish consumption in the context of public health and ecosystem services. 

Regarding the fish populations at Gwangyang Bay, we could roughly assess the abundance and dominance of fish populations in response to the spatial clustering groups determined by SOM. In terms of individual and species numbers, sites 8 and 11 appear to show higher productivity than the others (Table 2). Notably, this pattern was in accordance with SOM characterization (Group 1: sites 8 and 11). Considering the highest richness and biodiversity (*H′* = 2.35, Table 2), we assert that the high productivity at this site is closely associated with the significant influx of nutrients from the river. As for the nutrients, recent biochemical research from Gwangyang Bay has reported that the influx of phosphorus and nitrogen from rivers is highly correlated with the protein compositions in the ocean [61]. The same study underlined that the high proportion of protein under the abundant nutrient condition is related to the production of phytoplankton. However, the fish biomass and composition were slightly heterogeneous between the two sites (Table 2). In particular, total biomass of fish highlights the fact that site 8 is the most productive among the four sites (Table 3). While the SOM characterizes it as being part of the main channel (SOM Group 1), site 8 appears to be affected by abundant planktonic food associated with nutrient input from the river (SOM Group 3). Given this mixture pattern in site 8, the current study facilitates the need for additional monitoring and intense sampling near the river.

On the other hand, we draw attention to benthic food items for fish among the study sites. Interestingly, we found spatial variation of the biological entities, such as fish, shrimps, and crabs, across the four sites (Table 3). Seemingly, the fish communities are similar between site 8 and site 11 in terms of richness and diversity (Table 2). Nonetheless, we found that the benthic communities, such as shrimps and crabs, showed different habitat selections; the total biomass of shrimps was similar between the two sites, while that of crabs was significantly different (Table 3). It is remarkable that this pattern was opposite for the individual biomass between the two. Without direct evidence to explain the reason behind this pattern, however, knowing that marine benthic populations represent over 80% of primary food sources for fish [62,63], it is presumable that the spatial differentiation in benthic communities can lead to the abundance and composition of fish communities. Similarly, the distinct pattern of dominant fish populations was remarkable at sites 11 and 12, which are geographically very close to each other; while *Leiognathus nuchalis* were predominant (45%) at site 11, *Argyrosomus argentatus* were predominant (53.4%) at site 12 (Table 2). Interestingly, the biomass pattern of the shrimps differed from each other; site 11 showed the largest individual biomass, while site 12 showed the largest total biomass. This implies that there are more food resources available for specific fish populations (e.g., *Argyrosomus argentatus*) at site 12. At site 11, availability of the food sources might be limited in terms of both quantity and size (i.e., too large to eat; see Table 3). Lastly, within the outer bay, the richness and composition of the fish populations differed between sites 12 and 13. As previously discussed, this might also be related to the double-to-triple total biomass of shrimps at site 12 compared to the other three sites (Table 3). However, there is a great deal of uncertainty in investigating fish populations, given their variation in both space and time. Moreover, it is difficult and subjective to determine the spatial variation of fish populations, since they migrate from space to space. As a consequence, geographical distance may not be meaningful for determining the spatial characterization of ecological features in coastal areas, and more intensive surveys are required. Therefore, the spatial clustering by SOM can be an effective means to guide sampling and monitoring sites at Gwangyang Bay because the SOM groups seem to reflect biological differentiation in response to ambient physicochemical conditions.

## 5. Conclusions

To recap, the present environmental conditions of Gwangyang Bay were spatially characterized according to four different groups of water quality and sediment contamination conditions. From the results, we place emphasis on several interesting points: (i) the SOM manifests the dominant physicochemical attributes of each geographical zone associated with the patterns of water quality and sediment contamination; (ii) fish populations appear to be closely associated with their food sources (e.g., shrimps and crabs) as well as the ambient physicochemical conditions; and (iii) in the context of public health and ecosystem services, the SOM result can potentially offer guidance for fish consumption associated with sediment heavy metal contamination. The present study has scientific novelty in view of the expansion of SOM applications for understanding biological communities in association with their ambient physicochemical conditions. At the same time, however, we recognize limitations in representing general features of Gwangyang Bay, since the snapshot of our survey is limited in its ability to characterize a complex ecosystem. In this regard, consistent sampling and high-frequency investigations are needed to capture spatial variation and to delineate the temporal dynamics of water quality, sediment contamination, and fish populations. Nevertheless, our study highlights that the use of SOM is helpful for a first approximation of spatial heterogeneity in limno-oceanographic patterns for effective management of fisheries resources. 

## Figures and Tables

**Figure 1 ijerph-16-00409-f001:**
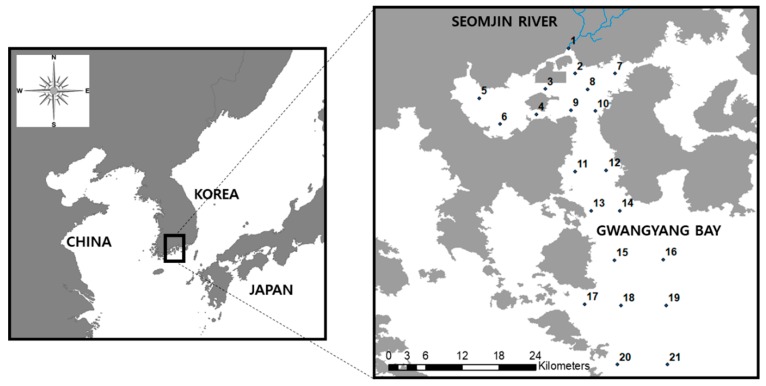
Map of the study sites (*N* = 21) at Gwangyang Bay in Korea.

**Figure 2 ijerph-16-00409-f002:**
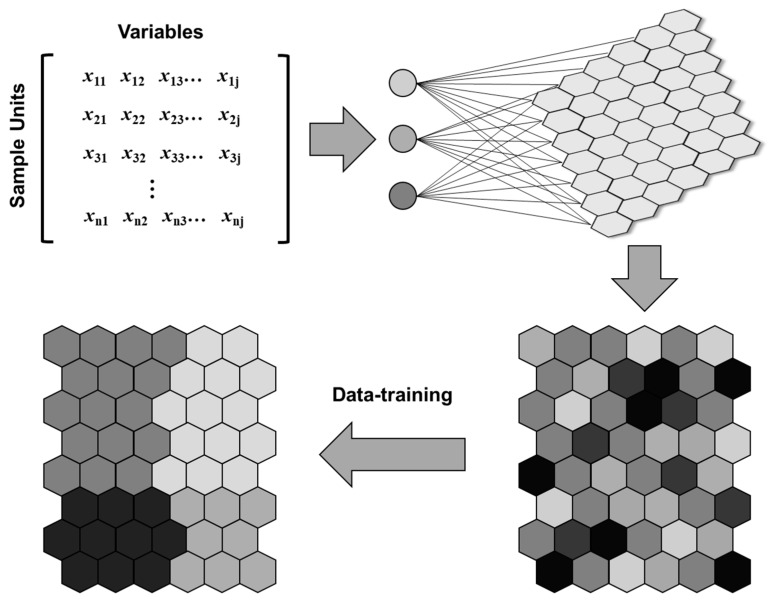
Conceptual structure of pattern recognition using Self-Organizing Map.

**Figure 3 ijerph-16-00409-f003:**
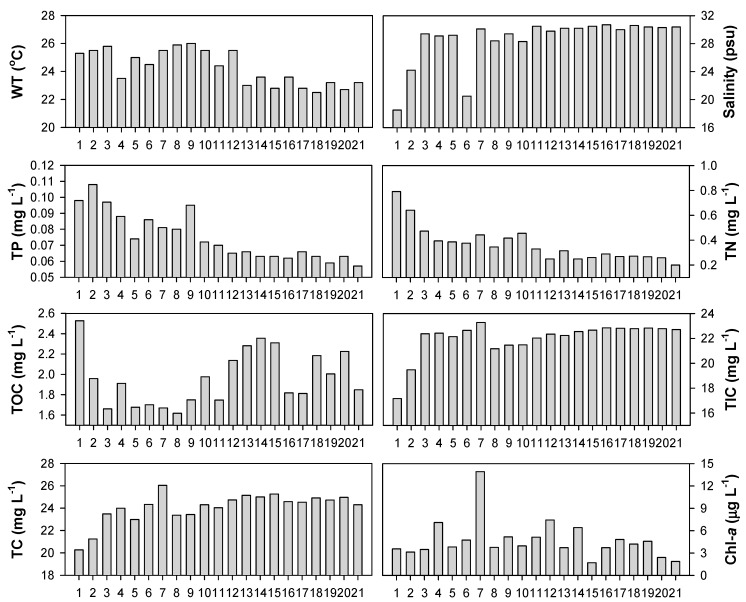
Water quality observation at Gwangyang Bay. The numbers on the x axis indicate the study sites.

**Figure 4 ijerph-16-00409-f004:**
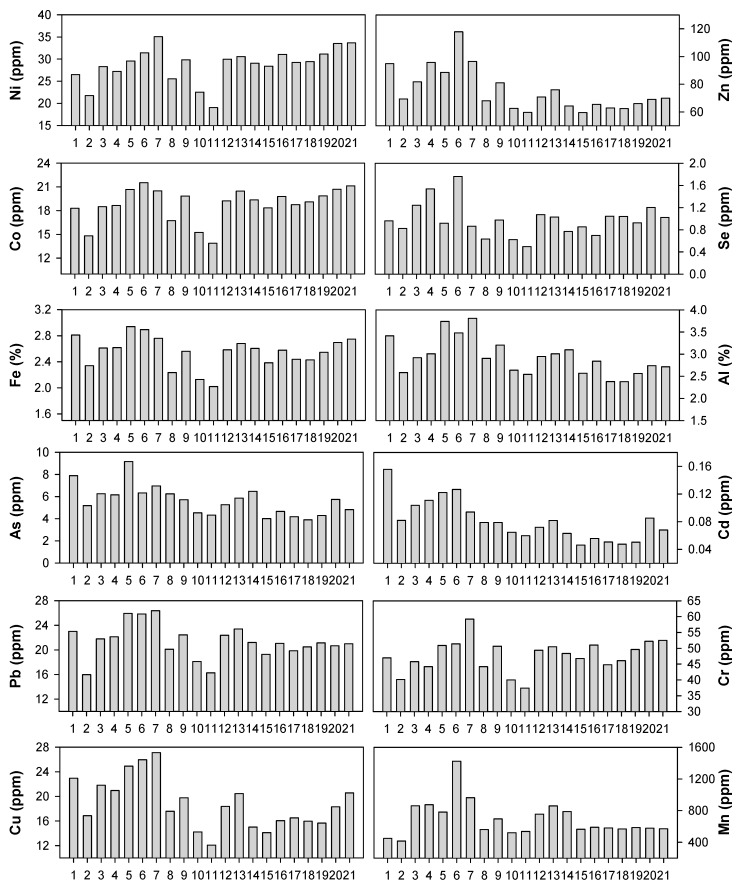
Sediment contaminant observation at Gwangyang Bay. The numbers on the x axis indicate the study sites.

**Figure 5 ijerph-16-00409-f005:**
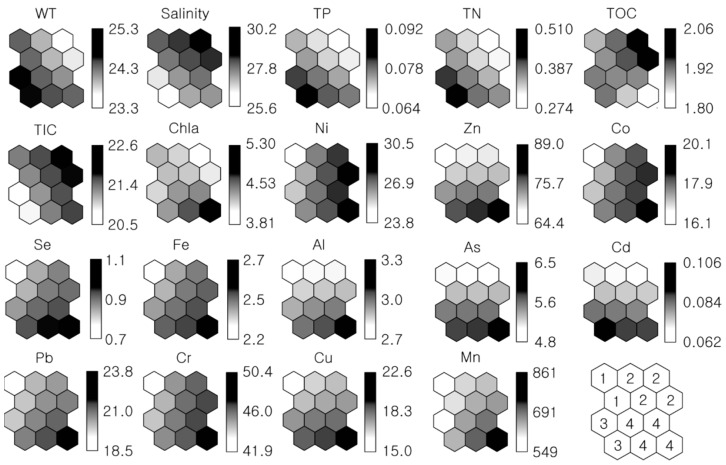
Self-Organizing Maps (SOMs) of water quality and sediment contaminants. The bars indicate the range of values (refer to Table 1 for the units). The right-bottom map presents four SOM clusters/groups.

**Figure 6 ijerph-16-00409-f006:**
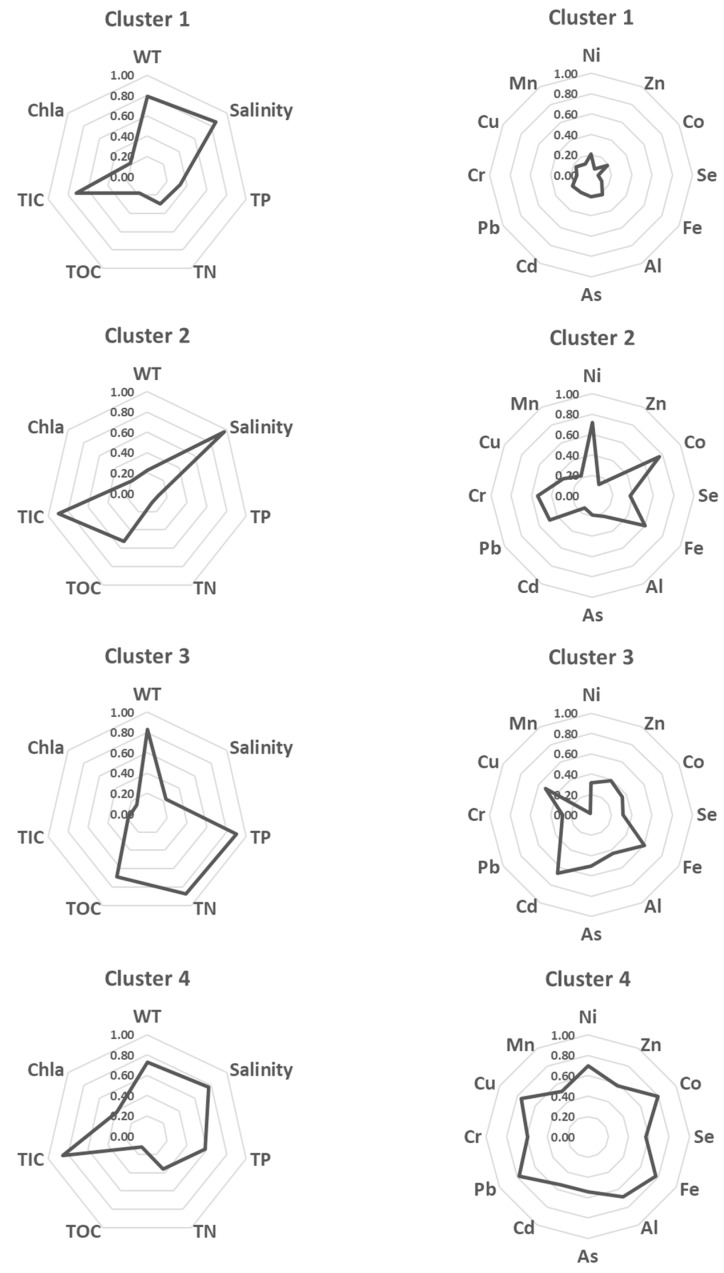
Comparison of water quality (left) and sediment contaminants (right) based on the SOM clusters. The values between 0 and 1 are normalized.

**Table 1 ijerph-16-00409-t001:** Characterization of water quality and sediment contaminants using SOM.

Variable	Unit	Mean	Group 1	Group 2	Group 3	Group 4
WT	°C	24.3	25.3	23.3	**25.4** *	25.1
Salinity	psu	28.6	29.1	**30.3** *	21.4	28.0
TP	mg L^−1^	0.075	0.074	0.063	**0.103** *	0.087
TN	mg L^−1^	0.365	0.377	0.262	**0.716** *	0.414
TOC	mg L^−1^	1.96	1.78	2.10	**2.24** *	1.73
TIC	mg L^−1^	22.0	21.6	**22.7** *	18.3	22.4
Chl-*a*	μg L^−1^	4.70	4.28	4.09	3.35	**6.37** *
Ni	ppm	28.7	22.4	**30.6** *	24.1	30.2
Zn	ppm	75.3	63.4	66.6	82.1	**93.5** *
Co	ppm	18.8	15.3	19.7	16.6	**20.0** *
Se	ppm	1.0	0.6	1.0	0.9	**1.2** *
Fe	%	2.6	2.1	2.6	2.6	**2.7** *
Al	%	2.9	2.7	2.7	3.0	**3.4** *
As	ppm	5.6	5.0	4.9	6.5	**6.8** *
Cd	ppm	0.08	0.06	0.06	**0.12** *	0.10
Pb	ppm	21.4	18.2	21.0	19.5	**24.1** *
Cr	ppm	47.7	40.5	49.1	43.6	**50.3** *
Cu	ppm	18.8	14.6	17.1	19.9	**23.4** *
Mn	ppm	691.3	538.5	643.6	431.9	**933.6** *

* Bold numbers indicate the highest mean value among all groups for each variable.

**Table 2 ijerph-16-00409-t002:** Observed fish populations at Gwangyang Bay in July 2018.

Common Name	Scientific Name	Site 8 (%) *N* = 556	Site 11 (%) *N* = 369	Site 12 (%) *N* = 204	Site 13 (%) *N* = 222	Sum *N* = 1351
Blackhead seabream	*Acanthopagrus schlegelii*	1 (0.2)				1
Cardinalfish	*Apogon lineatus*	13 (2.3)			23 (10.4)	36
Common skate	*Okamejei kenojei*	14 (2.5)	6 (1.6)	2 (1.0)		22
Conger eel	*Conger myriaster*		1 (0.3)			1
Daggertooth pike conger	*Muraenesox cinereus*	3 (0.5)	4 (1.1)	2 (1.0)		9
Dorsal soft ray	*Pampus echinogaster*	27 (4.9)	3 (0.8)	4 (2.0)	8 (3.6)	42
Dotted gizzard shad	*Konosirus punctatus*	1 (0.2)				1
False kelpfish	*Sebastiscus marmoratus*		15 (4.1)	1 (0.5)		16
Filamentous shrimpgoby	*Myersina filifer*			1 (0.5)		1
Fingerling rockfish	*Sebastes inermis*		1 (0.3)			1
Goblinfish	*Inimicus japonicus*	7 (1.3)	2 (0.5)			9
Grey stingfish	*Minous monodactylus*	1 (0.2)				1
Largehead hairtail	*Trichiurus lepturus*	4 (0.7)	5 (1.4)	4 (2.0)	1 (0.5)	14
Pinkgray goby	*Amblychaeturichthys hexanema*		12 (3.3)			12
Red barracuda	*Sphyraena* spp.				2 (0.9)	2
Red eel goby	*Ctenotrypauchen microcephalus*		5 (1.4)	1 (0.5)		6
Red tongue sole	*Cynoglossus joyneri*	12 (2.2)	13 (3.5)	3 (1.5)	1 (0.5)	29
Sand smelt	*Sillago sihama*	17 (3.1)				17
Scad	*Decapterus maruadsi*		2 (0.5)			2
Spiny red gurnard	*Chelidonichthys spinosus*	9 (1.6)	6 (1.6)	19 (9.3)	2 (0.9)	36
Spotnape ponyfish	*Leiognathus nuchalis*	263 (47.3)	166 (45.0)	22 (10.8)	94 (42.3)	545
Tidepool gunnel	*Pholis nebulosa*			2 (1.0)		2
White croaker	*Argyrosomus argentatus*	142 (25.5)	97 (26.3)	109 (53.4)	58 (26.1)	406
Wild marbled sole	*Pleuronectes yokohamae*	1 (0.2)				1
Yellow croaker	*Larimichthys polyactis*	41 (7.4)	31 (8.4)	33 (16.2)	32 (14.4)	137
No. of species		16	16	14	10	
Shannon diversity (*H′*)		2.35	1.97	1.94	1.44	

**Table 3 ijerph-16-00409-t003:** Spatial comparison in the biomass of fish, shrimps, and crabs at Gwangyang Bay in July 2018. Bold numbers indicate the highest value among the four sites.

Fauna Type	Biomass	Site 8	Site 11	Site 12	Site 13
		*N* = 556	*N* = 369	*N* = 204	*N* = 222
Fish	Individual biomass (g)	28.9	33.6	**49.0** *	25.6
	Total biomass (kg)	**16.0** *	12.4	10.0	5.7
Shrimps	Individual biomass (g)	8.40	**28.69** *	7.50	9.63
	Total biomass (kg)	0.42	0.83	**2.33** *	0.09
Crabs	Individual biomass (g)	4.39	**4.58** *	4.52	4.32
	Total biomass (kg)	**2.52** *	0.80	0.50	0.51

* Bold numbers indicate the highest mean value among all groups for each variable.

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
