# Peer review of "Explicit Characterization of Spatial Heterogeneity Based on Water Quality, Sediment Contamination, and Ichthyofauna in a Riverine-to-Coastal Zone"

_ijerph, 2019, doi:10.3390/ijerph16030409_

Round 1

Reviewer 1 Report

This paper concentrates on identifying the spatial pattern of water quality and sediment conditions at Gwangyang Bay Korea using unsupervised neural networks named Self-Organizing Map (SOM). A spatial correlation was found between water quality parameters and heavy metal contaminants. From their temporal analysis, they conclude that a spatial pattern of water quality parameters and heavy metal contaminants are determined by the source and the physical factors (i.e. river and currents). Their modeling method is relatively advanced compared to multivariate methods such as PCA or CCA. While this result is not a surprising finding, the approach and method used may be valuable to be published.

Several suggestions and comments are highlighted below:

1.    The English needs to be improved. There are grammatical errors found in the manuscript.

2.    Line 88: what is “apt”?  

3.    Line 102: What depth are those water temperature and salinity measured? And how these physical parameters change with temporal and spatial variation at the study area.

4.    Line 110: Change the word ‘using’ by ‘at’

5.    Line 169-173: The salinity pattern here is described only by looking at horizontal view of the study area but not vertically. Since the seawater stratification (i.e. fresh water on the top, saltier water at the bottom) may occur near the river mouth, this description is relatively weak.   

6.    Line 233: What is the difference between group and cluster? Should the subtitle in those rose plots write as group?    

7.    Line 234: Should be ‘Figure 6’ instead

8.    Line 262: Change “been increasing” by “increased”

9.    Line 272: Incorrect words “accounting for”

10.  Line 297-298: Is this currents analysis based on your study or others? If others, please provide the reference(s).

11.  Line 301: Better to use “acceptance condition” rather than “in good condition”.

12.  Line 303-308: Tigris River in Turkey has different physical and anthropogenic conditions. This information is not appropriate to be discussed here.   

13.  Line 323: What is “notwithstanding”?

Author Response

Reviewer 1:

This paper concentrates on identifying the spatial pattern of water quality and sediment conditions at Gwangyang Bay Korea using unsupervised neural networks named Self-Organizing Map (SOM). A spatial correlation was found between water quality parameters and heavy metal contaminants. From their temporal analysis, they conclude that a spatial pattern of water quality parameters and heavy metal contaminants are determined by the source and the physical factors (i.e. river and currents). Their modeling method is relatively advanced compared to multivariate methods such as PCA or CCA. While this result is not a surprising finding, the approach and method used may be valuable to be published.

Response: We deeply appreciate the valuable comments on our study, and we also respect the reviewer’s recommendation that the approach and method should be more highlighted. Following the comments, we revised the manuscript to be more focused to highlight how to rapidly and rigorously assess spatial characterization of ecological features. We also recognize that our current work lacks in characterizing physicochemical features of the bay in relation to seasonal variation and vertical variation. We stress, nevertheless, that our SOM analysis helps us establish additional monitoring sites and determine necessarily observed parameters/variables. In this context, as we mentioned in the introduction and discussion, this SOM approach is helpful as a first approximation of an environmental assessment for further in-depth studies in the same area. We finally decided to modify the title of manuscript to better emphasize the applicability of SOM to characterization of ecological features as a first approximation when the survey was temporally limited but spatially extensive with respect to coastal ecosystems. The modified title of our manuscript is “Explicit characterization of spatial heterogeneity based on water quality, sediment contamination, and ichthyofauna in a riverine-to-coastal zone”. Bearing this title in mind, we strived to re-construct the text, articulating methodological details. This comment was very helpful. Thank you!

Several suggestions and comments are highlighted below:

The English needs to be improved. There are grammatical errors found in the manuscript.

Response: We have been very careful of writing English, in order to properly convey the appealing points of our work, given that all the authors are not native speakers. With our concerns in mind, the revised manuscript was edited by a professional English editorial company.

Line 88: what is “apt”?  

Response: The sentence has been revised. “is apt to increase primary productivity as well as biological diversity” à “remains highly productive as well as biologically diverse”

Line 102: What depth are those water temperature and salinity measured? And how these physical parameters change with temporal and spatial variation at the study area.

Response: We thought that the previous methodological description caused some confusions to readers/reviewers. Thus, we strived to improve a resolution of methodological details, articulating sampling procedure and analytical protocols. Please see the section 2.2.

Line 110: Change the word ‘using’ by ‘at’

Response: Thank you for the suggestion. We replaced ‘using’ with ‘according to’.

Line 169-173: The salinity pattern here is described only by looking at horizontal view of the study area but not vertically. Since the seawater stratification (i.e. fresh water on the top, saltier water at the bottom) may occur near the river mouth, this description is relatively weak.   

Response: Thank you for the valuable comment on vertical saline gradient. We absolutely agree to this opinion. In addition to the saline gradient, we also believe that thermal stratification is on par with this comment. However, the vertical measurement of water quality was missing information in our exploratory survey as a first. The overarching goal of our study is to spatially evaluate the state of biological system in response to ambient physicochemical conditions. The data were not collected in finer resolution (in both time and vertical scale). In this regard, we firstly strived to characterize the spatial pattern in biogeochemical features in the bay. We could identify the main characteristics of Gwangyang Bay, and believe that this result will be able to help us further investigate more effectively. In fact, we plan to reduce the monitoring sites. Instead, more intense monitoring and sampling may be feasible.

Line 233: What is the difference between group and cluster? Should the subtitle in those rose plots write as group?    

Response: We modified the caption of Figure 5. The right-bottom map presents four SOM clusters/groups.

Line 234: Should be ‘Figure 6’ instead

Response: Sorry for the mistake. Figure 5 was corrected to Figure 6.

Line 262: Change “been increasing” by “increased”

Response: Sorry for the mistake. The sentence was correctly revised; ‘has been gradually been increasing’ à ‘has gradually been increased’

Line 272: Incorrect words “accounting for”

Response: We replaced ‘accounting for’ with ‘mapping’.

Line 297-298: Is this currents analysis based on your study or others? If others, please provide the reference(s).

Response: Thank you for the comment. The reference was inserted.

Line 301: Better to use “acceptance condition” rather than “in good condition”.

Response: The word was replaced according to the reviewer’s recommendation.

Line 303-308: Tigris River in Turkey has different physical and anthropogenic conditions. This information is not appropriate to be discussed here.   

Response: Thank you for letting us revisiting the context of our statement. We reconstructed the paragraph, thinking about logical flow and its transition. Several related literatures were added. à Counter to this judgement, the inner bay could be considered contaminated based on the concentrations of Zn, Pb, Cr, Cu, and Mn, using different criteria such as the Marine Sediment Pollution Index [50]. Several studies assessed heavy metal concentrations of sediment, benthos, and fish in river and estuary [51, 52]. Particularly in Europe, the Tigris River’s sediments showed twice the heavy metal concentrations found at Gwangyang Bay [12]. The same study reported that certain levels of heavy metals were detected in some fish species. Specifically, Silirus triostegus, Mastacembelus simacks, and Mystus halepensis accumulated heavy metals in their tissues containing liver, gill, and muscle (Cu: 2–5%, Ni: 0.6–1.0%, Mn: 0.2–1.6%, Fe: 0.1–0.7%; % concentration = mg kg-1 in wet weight / mg kg-1 in dry weight; Tables III–VI by Karadede-Akin and Unlu, [12]). In this context, there have been concerns about public health in relation to heavy metal contamination of sediment. Specifically, Yi et al. [52] reported that human health can be potentially influenced by eating fish that were affected by their benthic food sources and sediment contamination with respect to the lower Yangtze River. Thinking along the same lines, we presume that a certain amount of heavy metal could be transported from sediment to fish via benthic food-chain flow at Gwangyang Bay. The characterization of sediment contamination using SOM gives the information that intense monitoring and bio-assessment are further required for the inner bay. If commercial fish were exposed to the heavy metal contamination (even though it is below acceptable ranges of contamination), the advisory criteria for fisheries resources would be needed. To this end, the SOM result can offer guidance for fish consumption in the context of public health and ecosystem services.

Line 323: What is “notwithstanding”?

Response: We replaced it with ‘despite’.

Reviewer 2 Report

Comments to the Author

Reviewer recommendation and comments for Manuscript ID: ijerph-388518, entitled "Spatial analysis of water quality, sediment contamination, and ichthyofauna for integrated research of aquatic ecosystems pertaining to riverine-to-coastal zones"

General Comments:

This study claims to present the spatial analysis of water quality and sediment quality parameters over the Gwangyang Bay area but, I would say that I have not seen any such spatial analysis except a Self Organizing Map (SOM). Authors have used only one survey for water sampling which is not enough, it might be the case that the sediments/chemicals concentration was exceptionally high/low? Also, using data from only one survey will not reflect the seasonal variability of the sediments/chemicals, present in the studied area. Authors have not properly mentioned the instruments used and the methods used for collected samples. For me, this study is just a statistical analysis of the one month water quality data captured over the study area. Further, the authors have missed the linkage between the current study and to the environment and public health. I would not recommend its publication in IJERPH, with my further comments as below,

Major Comments:

1.     Page 2 line 53, I would suggest adding some recent references.

2.     Page 2 line 74, the application of SOM is no more novel, it has been used extensively in literature. please see the search results from Google Scholar; https://scholar.google.com.ph/scholar?hl=en&as_sdt=0%2C5&q=application+of+self+organizing+map+in+water&btnG=

3.     Page 2 line 88, what’s "apt"?

4.     Page 3 line 100, please provide a complete reference to this sampler.

5.     Page 3 lines 103-104, what was the method used for analyzing the nutrients and chlorophyll-a?

6.     Page 3 line 105, please name and provide the model of the UV spectrophotometer.

7.     Page 3 line 106, So, which chlorophyll-a concentration authors have used, the one analyzed in the lab or the one measured using a UV spectrophotometer? Confusing statement.

8.     Page 3 lines 112-114, why?

9.     Page 4 Figure 2, this figure is not necessary.

10.  Page 4 equations 1 and 2, references to equations 1 and 2 are missing.

11.  Page 4 line 154, what are those seven water quality variables?

12.  Page 5 Figure 3, why conceptual? not the actual?

13.  Page 5 line 164, the sampling period is very small i.e. only one month, which may have greater chances of exceptions.

14.  Page 5 lines 171-172, I don’t think that it’s a finding by the authors, it is well known that the normal range of salinity in sea is ppt.

15.  Page 5 line 173, figure 3 do not show the spatial pattern of phosphorus or any other water quality parameter!

16.  Page 7 Figure 5, the color bar (a variation between white and black) is not the actual depiction of the colors visible in the figures i.e. the greenish gray color variations are not available in the bar.

17.  Page 7 line 216, there are two Figures with name "Figure 5" which one you are referring to?

18.  Page 7 line 220, This is not only specific to the Gwangyang Bay, it is a well-known fact for any location on the earth.

19.  Page 10 Table 2, what is S8, S11, S12 and S13? there is no mention about it.

20.  Page 13 lines 362-368, this section has been ignored.

Author Response

Reviewer 2:

Reviewer recommendation and comments for Manuscript ID: ijerph-388518, entitled "Spatial analysis of water quality, sediment contamination, and ichthyofauna for integrated research of aquatic ecosystems pertaining to riverine-to-coastal zones"

General Comments:

This study claims to present the spatial analysis of water quality and sediment quality parameters over the Gwangyang Bay area but, I would say that I have not seen any such spatial analysis except a Self Organizing Map (SOM). Authors have used only one survey for water sampling which is not enough, it might be the case that the sediments/chemicals concentration was exceptionally high/low? Also, using data from only one survey will not reflect the seasonal variability of the sediments/chemicals, present in the studied area. Authors have not properly mentioned the instruments used and the methods used for collected samples. For me, this study is just a statistical analysis of the one month water quality data captured over the study area. Further, the authors have missed the linkage between the current study and to the environment and public health. I would not recommend its publication in IJERPH, with my further comments as below,

Response: We deeply appreciate the valuable comments as well as constructive criticism. First of all, we admitted that, given environmental seasonality, our study is limited to understanding the ecosystem in temporal scale. The overarching goal of our study (as a preliminary and exploratory survey) is to spatially evaluate the state of biological system in response to ambient physicochemical conditions. The data were not collected in finer resolution (in both time and vertical scale). In this regard, we firstly strived to characterize the spatial pattern in biogeochemical features in the bay. Certainly, our analysis might downplay biogeochemical responses to time-specific events. However, we have focused on identifying the main characteristics of Gwangyang Bay in space, and believe that this result will be able to help us further investigate more effectively. In fact, we plan to reduce the monitoring sites. Instead, more intense monitoring and sampling may be feasible. With these concerns in mind, the text has been heavily reconstructed. We also did our best to provide elaborate descriptions of methodology by adding more references. Thank you very much for this constructive feedback!

Major Comments:

Page 2 line 53, I would suggest adding some recent references.

Response: Yes, we added more recent references.

Page 2 line 74, the application of SOM is no more novel, it has been used extensively in literature. please see the search results from Google Scholar; https://scholar.google.com.ph/scholar?hl=en&as_sdt=0%2C5&q=application+of+self+organizing+map+in+water&btnG=

Response: We deleted ‘novel’.

Page 2 line 88, what’s "apt"?

Response: The sentence has been revised. ‘is apt to increase…’ à ‘remains highly productive as well as biologically diverse’

Page 3 line 100, please provide a complete reference to this sampler.

Response: Thanks for the comments. The section 2.2 has been rewritten, and we tried to provide elaborate methodological details as much as we could. Please see the section 2.2 in yellow highlight.

Page 3 lines 103-104, what was the method used for analyzing the nutrients and chlorophyll-a?

Response: Following the previous comment, the method section was reconstructed. We added the exact references for the standard methods used in this study.

Page 3 line 105, please name and provide the model of the UV spectrophotometer.

Response: Yes, we did it.

Page 3 line 106, So, which chlorophyll-a concentration authors have used, the one analyzed in the lab or the one measured using a UV spectrophotometer? Confusing statement.

Response: The method section has been reorganized. To avoid confusion, we simplified the text but added the related references.

Page 3 lines 112-114, why?

Response: The explanation was added in the revised method section.

Page 4 Figure 2, this figure is not necessary.

Response: Thank you for the comment, but would like to say that every single reader has different levels of background knowledge on SOM. The reviewer may have sufficient knowledge and experiences with SOM. However, many readers may prefer the conceptual figure to the mathematical formula. We appreciate if you consider our concerns.

Page 4 equations 1 and 2, references to equations 1 and 2 are missing.

Response: We deleted the equation numbers.

Page 4 line 154, what are those seven water quality variables?

Response: Thank you. We added the variables. à ‘The SOM was trained using nineteen input variables including seven water quality parameters (water temperature, salinity, total phosphorus, total, nitrogen, total organic carbon, total inorganic carbon, and chlorophyll a) and twelve sediment contaminants (Ni, Zn, Co, Se, Fe, Al, As, Cd, Pb, Cr, Cu, and Mn)’

Page 5 Figure 3, why conceptual? not the actual?

Response: Sorry for the mistake. We delete ‘conceptual’.

Page 5 line 164, the sampling period is very small i.e. only one month, which may have greater chances of exceptions.

Response: We think that this comment is on par with the general comment at the beginning. Please see our previous response. Thank you!

Page 5 lines 171-172, I don’t think that it’s a finding by the authors, it is well known that the normal range of salinity in sea is ppt.

Response: We would intend to highlight it as a finding, but described the spatial gradient of salinity across the bay. We also notice that those values (15 psu and 29 psu) are in normal range of seawater. However, given that 35 psu is a common value of salinity in ocean. Our opinion is that the values are still informative to delineate the saline gradient in space. Now we replaced ‘ppt’ with ‘psu’, as the other reviewer recommended ‘psu’.

Page 5 line 173, figure 3 do not show the spatial pattern of phosphorus or any other water quality parameter!

Response: We reworded the sentence. ‘spatial pattern’ à ‘distributional pattern’

Page 7 Figure 5, the color bar (a variation between white and black) is not the actual depiction of the colors visible in the figures i.e. the greenish gray color variations are not available in the bar.

Response: Thank you for the detailed comments. We retrofitted the contrast of the color in the figure.

Page 7 line 216, there are two Figures with name "Figure 5" which one you are referring to?

Response: Sorry! That is my mistake. The radical plots are named Figure 6.

Page 7 line 220, This is not only specific to the Gwangyang Bay, it is a well-known fact for any location on the earth.

Response: Thank you! The sentence was revised; ‘At Gwangyang Bay’ à ‘At Gwangyang Bay in common with typical coastal ecosystems’

Page 10 Table 2, what is S8, S11, S12 and S13? there is no mention about it.

Response: We corrected them to ‘Site 8, Site 11, Site 12, Site 13’.

Page 13 lines 362-368, this section has been ignored.

Response: Thank you for the comment. We filled the information about author contribution.

Reviewer 3 Report

Please, find my comments in the attached file

Author Response

Reviewer 3:

The authors present a data set of water and sediment quality parameters on Gwangyang Bay, and aim at relating them with fisheries resources (fishes, shrimps and crabs) from a punctual survey on June 2018.  The main objective according to the authors is to identify the spatial pattern of water quality and sediment conditions. Using Self-Organizing Map 69 (SOM) which is a type of unsupervised artificial neural network. A novel nonlinear analysis, Self-Organizing Map, for the rigorous assessment of spatial patterns.

Sampling and data collection section needs more details. For example:

Line 99: surface water (approximately top 5m). Please, it is not clear if the authors sampled at 5m depth or integrated sample from the first 5 meters.  Specify the sampling method and instrument.

Response: We appreciate the valuable comment. In fact, the other two reviewers also gave similar comments on the methodological description. Accordingly, we revised heavily the method section, articulating the sampling and analytical details. Please see the section 2.2.

Line 104: avoid informal abbreviations such as “lab”

Response: We replaced it with ‘laboratory’ as a full name.

Line 105: Korean Ministry of Oceans and Fisheries: please, include reference

Response: Thank you! We provide two references for the protocol.

Line 110: 850 oC must be written without empty space 850oC, here and along the manuscript.

Response: It was corrected to ‘850℃’.

Line 112: sediment samples (0.5 g per sample). But sediment samples were taken with an Ekman sampler and the authors used only 0.5 g, please specify how do you separated this aliquot.

Response: As previously mentioned, we reconstructed the method section. We provided the information which the reviewer commented on. Please see the section 2.2. Thank you!

Line 117: the survey for ichthyofauna. Authors should provide references for their methodology.

Response: Same as the previous response, please see the section 2.2. We provided the information about every single point raised by the reviewer.

Application of machine learning to the data analysis section

As far as I know, SOMs can operate in two modes: training and mapping. I guess that the authors are operating with mapping.

Line 155: The size of the SOM was selected following the rule of… For me it is not clear what the authors mean. Is it cluster number?

Response: The sentence was reworded. à ‘The SOM size (i.e., the number of hexagons in the map) was determined based on the rule’.

Line 160: Figure 3. This Figure is part of the results, and should appear after naming it in the text.

Response: Figure 3 has been moved to the result section.

Results section

Line 171-172 seriously salinity is expressed in ppt? I think this must be a mistake. For salinity use PSU or nothing.

Response: We expressed the values in units of parts per thousand (ppt or ‰; so-called Knudsen salinity). However, we replaced them with the values in practical salinity unit (psu). They are not identical but nearly the same.

Lines 208 and following “In summary, there is spatial variation in the heavy metal contamination of sediment; however, it was quite difficult to quantitatively distinguish the characterization of sediment contamination in space, based merely on the visual comparison using bar graphs.” This is quite obvious. Before publishing authors should always do the effort to search the best way for explaining and showing their data. If Figure 3 and 4 are not useful don’t include them.

Response: Thank you for the valuable comment. We agree to the reviewer’s opinion that difficulty of comparison using bar graphs is quite obvious. This is the advantage of SOM application we highlight throughout the manuscript. We think that this comment basically stems from the similar concerns and confusions that the other reviewers also advised. Keeping these concerns in mind, we strived to revise the manuscript to be more focused to highlight how to rapidly and rigorously assess spatial characterization of ecological features. The current study presents the SOM application to characterization of biological responses to ambient physicochemical features. As the reviewer also recognizes difficulty of comparison using bar graphs, the SOM clusters represented different biogeochemical features at Gwangyang Bay by classifying four groups. In this regard, our opinion is that Figures 3 and 4 are good comparison against the SOM results. We stressed in the introduction and discussion that this SOM approach is helpful as a first approximation of an environmental assessment for further in-depth studies in the same area. To this end, we decided to modify the title of manuscript to better emphasize the applicability of SOM to characterization of ecological features as a first approximation when the survey was temporally limited but spatially extensive with respect to coastal ecosystems. The modified title of our manuscript is “Explicit characterization of spatial heterogeneity based on water quality, sediment contamination, and ichthyofauna in a riverine-to-coastal zone”.

Figure 5 is hard to understand. Acronyms are not defined in figure caption. In addition, authors do not explain the meaning of the cells or the numbers. Does it mean we have 4 clusters?  Which are the sampling sites linked to each cluster?

Response: Sorry for the missing information. We provided full names of each variables in the section 2.3. As the reviewer understands correctly, the number indicates each cluster in SOM. The corresponding sampling sites were described in the section 3.3; Group 1 represented the main channel of the bay (the sites 8, 10, and 11); Group 2 mostly belonged to the outer bay (the sites 12–21); Group 3 corresponded to the estuary (the sites 1 and 2); and Group 4 covered the inner bay (the sites 3–7, and 9).

Line 220 higher water temperatures corresponded to lower salinity, and vice versa. But in line 165 Overall, lower water temperatures were recorded in the outer bay than in the estuary and inner bay. Isn’t it contradictory?

Response: We believe that it is a correct statement. It is typical of physicochemical gradient in coastal areas. Thus those two statements are consistent.

Line 234 Is again Figure 5, I guess it is Figure 6. The order of figure 5, 6 and Table 1 makes very hard to understand the SOM analysis. Figure 6 is not exploited at all, can the authors analyse the relation between water and sediment for each cluster? I think that more statistical analysis is needed. Maybe SOM could be helpful for mapping, but it will be necessary to assess If there exist statistically significant differences among clusters. Table 1 talks about the “highest values” but that should be statistically assessed to prove if it is relevant.

Response: Sorry! By mistake, we denoted Figure 5 twice. One of the figures is corrected to Figure 6. Regarding the relationship between the variables for each cluster, the reviewer’s comment is brilliant. However, we confront the barrier of statistical power within each cluster. For example, the cluster 3 contains only two sites (1 and 2). Namely, a correlation coefficient cannot be obtained. Aside from the sample size, SOM inherently infers the relationship in a different way (linear versus nonlinear), compared to statistical methods. The bottom line is that statistical analysis will not be fully supportive to this result. However, we believe that this approach is acceptable when each cluster contains sufficient amount of data. We deeply appreciate this valuable comment.

Table 2 and Table 3 should be included as results, not in the discussion section. Results and discussion section are not well organized.

Response: In fact, we have thought about the contents for the result and discussion. However, given the main analysis of SOM using physicochemical data, we thought it is more desirable to discuss how to connect the SOM result to biological responses. The discuss section has been revised in consideration of the reviewer’s concern. Thank you!

Discussion

Discussion level is low. Some examples are:

Lines 300-303 “300 Nevertheless, the levels of sediment contamination mostly remained in a good condition based on the pollution index [11]. However, the concentrations of Zn, Pb, Cr, Cu, and Mn were considered contaminated based on the Marine Sediment Pollution Index [44].”  It is contradictory. In any case authors should explain that index.

Response: Thank you for the valuable comment. The other reviewer also commented similarly on this part. The paragraph has been reconstructed as follows. à ‘Given that heavy metal contaminants stem from exogenous sources of chemical pollutants (i.e., the industrial complex adjacent to the bay), we conjecture that the relatively fast current and hydrodynamics in the middle of the bay could reduce the quantity of heavy metal contaminants settling into the sediments [49]. On the contrary, the inner bay corresponded to high concentrations of heavy metals, which implies that the relatively enclosed shape of the bay could lead to the accumulation of these metals. Nevertheless, the levels of sediment contamination mostly remained in acceptable condition based on the pollution index [11]. Counter to this judgement, the inner bay could be considered contaminated based on the concentrations of Zn, Pb, Cr, Cu, and Mn, using different criteria such as the Marine Sediment Pollution Index [50]. Several studies assessed heavy metal concentrations of sediment, benthos, and fish in river and estuary [51, 52]. Particularly in Europe, the Tigris River’s sediments showed twice the heavy metal concentrations found at Gwangyang Bay [12].’

Lines 304-311 The authors summarized another study results which is not relevant at all. And based on that they affirm that “Despite no investigation of the heavy metal contamination in fish tissues, we suppose that a certain amount of heavy metal could be absorbed by eating the fish.” These are speculations.

Response: We deeply appreciate it. We acknowledge the elucidation of possible causes and effects based on the previous literature, but cannot admit that the explanations are irrelevant. We think that this comment may be derived from inappropriate expressions in our text. With this skepticism in mind, we revised the context carefully. à ‘there have been concerns about public health in relation to heavy metal contamination of sediment. Specifically, Yi et al. [52] reported that human health can be potentially influenced by eating fish that were affected by their benthic food sources and sediment contamination with respect to the lower Yangtze River. Thinking along the same lines, we presume that a certain amount of heavy metal could be transported from sediment to fish via benthic food-chain flow at Gwangyang Bay. The characterization of sediment contamination using SOM gives the information that intense monitoring and bio-assessment are further required for the inner bay. If commercial fish were exposed to the heavy metal contamination (even though it is below acceptable ranges of contamination), the advisory criteria for fisheries resources would be needed. To this end, the SOM result can offer guidance for fish consumption in the context of public health and ecosystem services.’

Lines 319-320 “Moreover, the influx of phosphorus and nitrogen from rivers is highly correlated with the supply of protein from the river to the ocean [45]” poor interpretation.

Response: Thank you for the comment. The paragraph was heavily reconstructed. à ‘As for the nutrients, the recent biochemical research from Gwangyang Bay has reported that the influx of phosphorus and nitrogen from rivers is highly correlated with the protein compositions in the ocean [53]. The same study underlined that high proportion of protein under the abundant nutrient condition is related to production of phytoplankton.’

Line 330 “knowing that marine benthic populations are the primary food sources for fish” but authors have not previously explained the fishes characteristics, for better understanding they should explain because planktonic population can be as important depending on fish species

Response: We inserted the related references. The references informed that >80% of food sources were shrimps. à ‘knowing that marine benthic populations occupy over 80% as primary food sources for fish [54, 55]’

Conclusions

Conclusions are obvious, for example “the river appears to play an important role in supplying nutrients (phosphorus and nitrogen) to the bay;” No need of this research for that result.

Response: We appreciate the comment. The conclusion was rewritten in response to this valuable comment.

In summary, I think that the authors should be a much bigger effort in obtaining more interesting results from their data. Exploit points of interest like relation between industry location and sediment pollution using SOM, try to explain temperature changes better, try to find if observed differences are relevant or not….

Response: The discussion and conclusion were reconstructed heavily. The revised parts were in yellow highlight.

We hope that our revised context is satisfactory and well-fitted to the reviewer’s requests. My co-authors and I deeply appreciate all the constructive feedback from the three reviewers.

Round 2

Reviewer 3 Report

I am truly sorry, but after the improved description of methodology section I have far more concerns.

I think that sampling metholodogy and choice of parameters are not scientifically adequate for the objectives of the paper.

The authors sampled 21 sites between 0 m and 5m using van Dorn horizontal sampler, and they do not have vertical profiles of salinity. But hey are sampling on the Seomjin River estuary, which according to authors has a an annual mean discharge of 2,298×106 m3 yr-1.

The first step should have been determining the influence of the river, and for that salinity profiles are essential. Sampling depth is a key parameter, sampling between 0 to 5 m is very imprecisse and variations can be due to different sampling depth, specially near the river mouth. But, in addition, authors present figure 1 with no scale, so we cannot even now the distance of sampling sites from river mouth. they base their descriptions on three areas: inner bay, outer bay and estuary. but these areas are not delimited anywhere, and site 6 shows a low salinity in spite of being further from river mouth. This is not explained. We do not have information of river discharge on June, not about currents or prevailing winds.

Also, authors analyses TP and TN (by the way, they do not include analysis methodology) in water samples. But for primary production, and consequently for fisheries, total nutrients are not relevant, but their dissolved forms, which are the one available for phytoplankton.

Regarding salinity units ppt is or can be parts per trilion (not part per thousand) that is the reason I asked about salinity values in my previous revision. Figure 3 still has ppt units.

So, due to mtehodology inconsistences I do not recommend the publication of this study.

Author Response

Reviewer’s comments:

(1) The authors sampled 21 sites between 0 m and 5m using van Dorn horizontal sampler, and they do not have vertical profiles of salinity. But hey are sampling on the Seomjin River estuary, which according to authors has an annual mean discharge of 2,298×106 m3 yr-1. The first step should have been determining the influence of the river, and for that salinity profiles are essential. Sampling depth is a key parameter, sampling between 0 to 5 m is very imprecisse and variations can be due to different sampling depth, specially near the river mouth.

Response: The bay receives an annual mean discharge of 2,298×106 m3 yr-1 from the Seomjin River [18]. Thus, the value was obtained from the previous study in the same area.

ü  Kang, C.-K.; Kim, J. B.; Lee, K.-S.; Kim, J. B.; Lee, P.-Y.; Hong, J.-S., Trophic importance of benthic microalgae to macrozoobenthos in coastal bay systems in Korea: dual stable C and N isotope analyses. Mar. Ecol. Prog. Ser. 2003, 259, 79-92.

We also possess the same opinion that sampling depth is critical in terms of analytical precision. However, we appreciate if the reviewer understands that our current study has focused on the combination of surface water quality and corresponding sediment in space. Concerning the surface water samples between 0 and 5m, we also acknowledge a certain level of vertical variation. Close to the river we were very careful of water sampling because the depth was relatively low. Similar to the reviewer’s concern, we collected the water sample between 0 and 1m, particularly near the river mouth. Hence, we added one more sentence in Line 103-104 (Particularly for sampling water adjacent to the river mouth, we sampled water at the depth between 0m and 1m due to shallow depth relative to ocean).

(2) But, in addition, authors present figure 1 with no scale, so we cannot even now the distance of sampling sites from river mouth.

Response: We inserted the scale bar into the map (Figure 1).

(3) they base their descriptions on three areas: inner bay, outer bay and estuary. but these areas are not delimited anywhere, and site 6 shows a low salinity in spite of being further from river mouth. This is not explained. We do not have information of river discharge on June, not about currents or prevailing winds.

Response: We would like to emphasize the aim of our study using a SOM clustering analysis which is known as a robust and powerful tool for nonlinear feature recognition in relation to data learning. As a result, the SOM clearly identified the spatial characteristics, which implies that the Gwangyang Bay could be characterized as four distinct features based on surface water quality and sediment contamination. Therefore, the spatial characterization determined by SOM is a good starting point for following studies as a first approximation of complex coastal ecosystems. To this end, the four distinct areas were based on the map (Figure 1), and we specified the corresponding sites to each clustering group.

In Line 253-256, Group 1 represented the main channel of the bay (the sites 8, 10, and 11); Group 2 mostly belonged to the outer bay (the sites 12–21); Group 3 corresponded to the estuary (the sites 1 and 2); and Group 4 covered the inner bay (the sites 3–7, and 9). The site 6 is part of Group 4. Although we also recognized lower salinity in the site 6, we note that SOM characterized this site similarly with other sites 3, 4, 5, 7 and 9. To our understanding, this is because the representative attribute of habitat/site cannot be determined by one factor or two. This is the robustness of the SOM applications to nonlinear feature selection in ecology.

(4) Also, authors analyses TP and TN (by the way, they do not include analysis methodology) in water samples. But for primary production, and consequently for fisheries, total nutrients are not relevant, but their dissolved forms, which are the one available for phytoplankton.

Response: The original sentence included the methods used for the nutrients that are also measured using a spectrophotometer. To clarify it, we modified the description as follows: The nutrient and chlorophyll-a concentrations were analyzed from the water samples in the laboratory. Those concentrations were based on spectrophotometry (Model: OPTIZEN POP Series UV-Vis). Particularly for chlorophyll-a measurement, the water samples were filtered through a 0.45 μm pore-size membrane (Model: Advantec MFS membrane filter); the membrane was then homogenized prior to the spectrophotometry (Model: OPTIZEN POP Series UV-Vis). Their concentrations were measured according to standard analytical methods proposed by the Korean Ministry of Oceans and Fisheries [21] and the Korean Ministry of Environment [22]. Organic and inorganic carbon concentrations were measured using a carbon analyzer (Model: vario TOC cub) according to 850℃ combustion catalytic oxidation methods. Regarding bioavailable nutrients, we think that total nutrients can be used as surrogates. There are several reasons in the study area; (i) phytoplankton biomass (4.7 μg Chla L-1) was fairly low; (ii) particulate nutrients were also low given the relationship between organic (1.96 mg C L-1) and inorganic forms (22.0 mg C L-1) (see Table 1); and (iii) in this case, total and dissolved nutrients have strong positive relationship. Therefore, we presume that total phosphorus and total nitrogen can also indicate bottom-up control.

(5) Regarding salinity units ppt is or can be parts per trilion (not part per thousand) that is the reason I asked about salinity values in my previous revision. Figure 3 still has ppt units.

So, due to mtehodology inconsistences I do not recommend the publication of this study.

Response: We corrected Figure 3.
